# CMRAG: Co-modality–based visual document retrieval and question answering

## Abstract

Retrieval-Augmented Generation (RAG) has become a core paradigm in document question answering tasks. However, existing methods have limitations when dealing with multimodal documents: one category of methods relies on layout analysis and text extraction, which can only utilize explicit text information and struggle to capture images or unstructured content; the other category treats document segmentation as visual input and directly passes it to visual language models (VLMs) for processing, yet it ignores the semantic advantages of text, leading to suboptimal retrieval and generation results. To address these research gaps, we propose Co-Modality–based RAG (**CMRAG**) framework, which can simultaneously leverage texts and images for more accurate retrieval and generation. Our framework includes two key components: (1) a Unified Encoding Model (**UEM**) that projects queries, parsed text, and images into a shared embedding space via triplet-based training, and (2) a Unified Co-Modality–informed Retrieval (**UCMR**) method that statistically normalizes similarity scores to effectively fuse cross-modal signals. To support research in this direction, we further construct and release a large-scale triplet dataset of (query, text, image) examples. Experiments demonstrate that our proposed framework consistently outperforms single-modality–based RAG in multiple visual document question-answering (VDQA) benchmarks. The findings of this paper show that integrating co-modality information into the RAG framework in a unified manner is an effective approach to improving the performance of complex VDQA systems.

## 1 Introduction

Large language models (LLMs) have received extensive attention in recent years (Touvron et al., 2023; Achiam et al., 2023; Guo et al., 2025; Yang et al., 2025a), but they have inherent limitations in handling out-of-domain knowledge (Ji et al., 2023). To address this issue, RAG integrates external knowledge retrieval with the generation process (Lewis et al., 2020; Guu et al., 2020; Karpukhin et al., 2020; Chen et al., 2025). RAG achieves wide success in open-domain question answering, knowledge retrieval, and dialogue systems, and becomes an effective means of extending the knowledge boundaries of LLMs (Ram et al., 2023; Gao et al., 2023). However, most external data sources (e.g., documents) are essentially multimodal (Jeong et al., 2025; Faysse et al., 2025), often containing natural language text, formulas, tables, images, and complex layout structures. How to effectively leverage such multimodal information in question answering remains a challenging problem that is not fully solved.

One line of approaches is text-based RAG, which typically relies on layout parsing and text extraction (Xu et al., 2020; Dong et al., 2025; Yang et al., 2025b; Perez & Vizcaino, 2024). These methods first detect document layouts and then extract textual information for subsequent retrieval and generation, as shown in Fig. 1(a). While stable at the semantic level, they struggle to handle content such as images and tables. Recently, VLMs (Ghosh et al., 2024; Radford et al., 2021; Alayrac et al., 2022) enable RAG systems to process documents directly as images (Faysse et al., 2025; Yu et al., 2025; Qi et al., 2024a; Wang et al., 2025b), giving rise to vision-based RAG (Huang et al., 2022; Kim et al., 2022; Yu et al., 2025). Specifically, as shown in Fig. 1(b), these methods divide document pages into image segments and perform retrieval and reasoning through visual understanding models. Although they capture non-textual information, they often overlook the precise information carried by text, leading to performance bottlenecks.

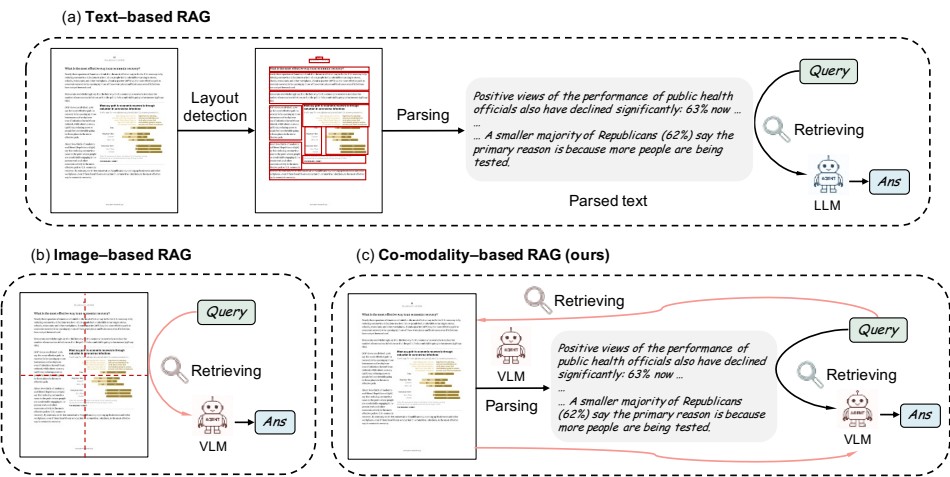

Figure 1: Comparison among (a) text–based RAG, (b) image–based RAG, and (c) co-modality–based RAG.

To overcome these limitations, we propose a novel co-modality–based RAG (**CMRAG**) framework, which unifies text and image modalities, as illustrated in Fig. 1(c). In this framework, we first parse documents to extract structured text and image segments. For a given query, we propose a novel retrieval pipeline named CMRAG-Retrieval (**CMRAG-R**) to retrieve both text and visual representations, ensuring that semantic matching from text and perceptual grounding from images are simultaneously leveraged. Finally, we feed the co-modality evidence into a VLM to integrate information and generate answers.

The CMRAG-R consists of two key components: (1) A unified encoding model (**UEM**) that projects queries, images, and parsed texts into a shared, comparable latent space. This model is trained using a triplet-based objective with a sigmoid loss to ensure robust alignment across modalities. (2) A unified co-modality-informed retrieval method (**UCMR**) that normalizes similarity scores to mitigate distributional discrepancies between modalities, enabling a more effective retrieval process. The main contributions of this paper are summarized as follows:

- We propose **CMRAG**, a novel co-modality–based RAG framework that leverages both text and image representations for significantly improved retrieval and generation on visual documents.

- We propose **UEM**, a unified encoding model that uses a single set of encoders for all modalities, which is trained end-to-end using a pairwise sigmoid loss on query-text and query-image triplets to create a unified embedding space.

- We propose **UCMR**, a unified co-modality-informed retrieval method that employs statistical normalization to effectively combine visual and textual similarity scores, addressing the inherent challenges of cross-modal score fusion.

- We parse and release **a large-scale triple dataset** constructed from an open-source visual document corpus, providing (query, image, text) triplets to facilitate future research in co-modality learning for the community.

- We conduct extensive experiments on multiple VDQA benchmarks, showing that our method consistently outperforms strong single-modality RAG baselines and demonstrates the effectiveness of co-modality integration.

## 2 RELATED WORK

In this section, we discuss the recent studies related to multi-modal RAG (MMRAG). Specifically, we primarily focus on two folds: (1) knowledge–based MMRAG and document–based MMRAG. We also review video–based MMRAG studies in Appendix C.

**Knowledge–based MMRAG** refers to retrieving knowledge (text or image modality) from external sources such as Wikipedia articles and websites to answer textual or visual queries (Talmor et al., 2021; Marino et al., 2019; Chang et al., 2022; Schwenk et al., 2022; Mensink et al., 2023; Chen et al., 2023; Ma et al., 2024a; Hu et al., 2025). Although the external knowledge database can enhance the system performance (Caffagni et al., 2024), the key issue of knowledge–based MMRAG is the inconsistency between textual and visual queries as well as the external knowledge database (Chen et al., 2022; Lin et al., 2023; Zhang et al., 2024b). To address this issue, (Lin & Byrne, 2022) adopted multiple algorithms, including object detection, image captioning, and OCR, to transform visual queries into language space, and proposed a joint training scheme to optimize retrieval and generation simultaneously. A similar training strategy was also used by (Adjali et al., 2024). Also, (Yan & Xie, 2024) used a consistent modality for both retrieval and generation: visual modality for retrieval (visual queries and Wikipedia article pages) and textual modality for generation (textual queries and wiki articles). A similar strategy can also be found in RORA (Qi et al., 2024a). In addition, (Tian et al., 2025) proposed cross-source knowledge reconciliation for MMRAG, which could address the inconsistency between textual and visual external knowledge.

**Document–based MMRAG** refers to retrieving a few document pages from one or multiple documents to help generate answers for given questions (Methani et al., 2020; Masry et al., 2022; Tanaka et al., 2023; Tito et al., 2023; Ma et al., 2024b; Li et al., 2024; Hui et al., 2024; Qi et al., 2024b; Cho et al., 2024; Wang et al., 2025a; Li et al., 2025; Wasserman et al., 2025; Faysse et al., 2025). Traditionally, documents were parsed using detection models (Ge et al., 2021) and OCR engines (Smith, 2007), and the extracted components (e.g., text) were input to LLMs to generate answers (Riedler & Langer, 2024; Faysse et al., 2025). With the proliferation of VLMs, a few studies (Faysse et al., 2025; Yu et al., 2025; Wang et al., 2025b) processed document pages as images directly. Specifically, they used VLMs to encode queries and document pages as text and images, respectively, based on which the similarity scores between queries and visual document pages can be calculated. This method paves the way for document–based MMRAG, as it does not need to parse documents and can retrieve document pages directly. However, this method overlooks text modality in documents, which may degrade the performance of RAG systems.

## 3 METHODOLOGY

In this section, we introduce the problem definition, the proposed CMRAG framework, the design and training method of the CMRAG-retrieval, and computational cost analysis.

### 3.1 PROBLEM DEFINITION

We formulate the task of Visual Document Question Answering (VDQA) within a Retrieval–Augmented Generation (RAG) framework. A collection of visual documents (e.g., PDFs, scanned articles) serves as the knowledge source for answering user queries. As illustrated in Fig. 2(a), each document page $p_i$ is first parsed by a Vision Language Model (VLM) $\mathcal{V}$ to extract its structured multimodal content. This parsing step produces both a visual representation $I_i$ (which represents the entire page image in this study) and a textual representation $T_i$, such that $I_i, T_i = \mathcal{V}(p_i)$. The complete set of all parsed pages constitutes the candidate evidence pool, denoted as $\mathcal{D} = \{\{I_1, T_1\}, \{I_2, T_2\}, \ldots, \{I_M, T_M\}\}$, where $M$ is the total number of pages. For a given query $q$, the objective of the retriever $\mathcal{R}$ is to identify the top-$k$ most relevant pages $P_k$ based on a multimodal similarity score $s(q, \{I_i, T_i\})$.

Considering recent advances in VLMs (Bai et al., 2025; Du et al., 2025) enable them to directly process multiple naive-resolution page images, the retrieved evidence $P_k$ is directly combined with the query $q$ into a structured prompt $\mathcal{P}(q, P_k)$, which is then fed into a generator model $\mathcal{G}$, typically a powerful VLM, to produce the final answer: $\hat{a} = \mathcal{G}(\mathcal{P}(q, P_k))$ (as shown in Fig. 2(d)). A comprehensive list of notations is provided in Appendix A. Also, we introduce all the prompt templates used in this study in Appendix B.

### 3.2 CMRAG-RETRIEVAL

In this section, we present a unified model architecture for VDQA, where parsed texts are long and semantically diverse. Existing image–text pretraining models, such as CLIP (Radford et al., 2021),

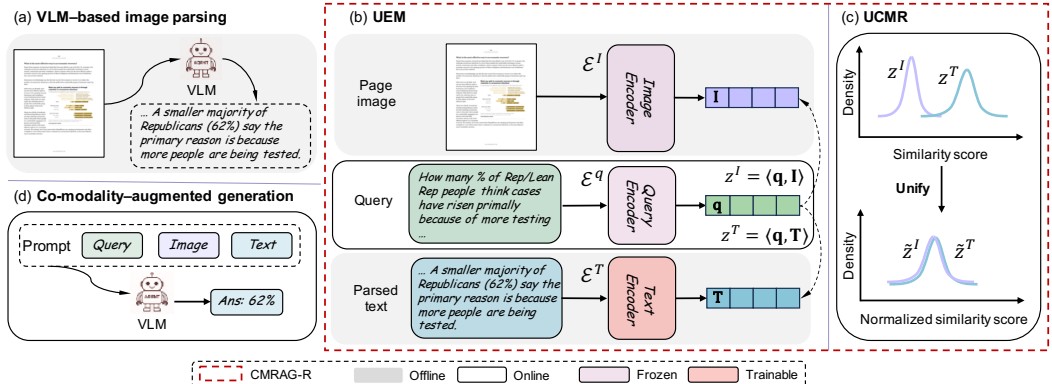

Figure 2: An overview of the proposed CMRAG framework. (a) A VLM is prompted to parse visual documents offline. (b) Images, parsed texts, and given queries are encoded uniformly in a shared space. Images and texts can be encoded and indexed offline to accelerate the online RAG systems. (c) The calculated similarity scores of visual and textual modalities are unified to a comparable distribution, ensuring a more accurate retrieval. (d) A VLM generator is prompted to generate the final answer based on the query and retrieved evidence.

SigLIP (Zhai et al., 2023) and SigLIP2 (Tschannen et al., 2025), are optimized for short texts and degrade on long documents. In addition, they remain biased toward strong image–text correlations in pretraining data. In VDQA, such correlations are sparse, as each image relates to only a small portion of the document. To address this, we propose the **CMRAG-R**, which jointly embeds queries, images, and parsed texts into a shared latent space for efficient document-level retrieval.

**Unified encoding model (UEM).** A streamlined yet expressive framework that jointly embeds queries, parsed document texts, and document images into a shared latent space, substantially simplifying the retrieval process. Built on the SigLIP backbone (Zhai et al., 2023), our design integrates three encoders—query $\mathcal{E}^q$, image $\mathcal{E}^I$, and text $\mathcal{E}^T$—into a unified architecture (Fig. 2(b)). We directly reuse the pretrained $\mathcal{E}^q$ and $\mathcal{E}^I$ from SigLIP to preserve the powerful multimodal alignment acquired at scale, and initialize $\mathcal{E}^T$ as a length-extended copy of $\mathcal{E}^q$ to accommodate long parsed texts. This yields query, image, and text embeddings:

$$\mathbf{q} = \mathcal{E}^q(q) \in \mathbb{R}^{1 \times d}, \quad \mathbf{I} = \mathcal{E}^I(I) \in \mathbb{R}^{1 \times d}, \quad \mathbf{T} = \mathcal{E}^T(T) \in \mathbb{R}^{1 \times d}, \tag{1}$$

where $d$ denotes the embedding dimension. A key benefit of UEM is that a single forward pass of $\mathcal{E}^q$ suffices to produce a query representation that can be directly compared against both $\mathbf{T}$ and $\mathbf{I}$, enabling efficient cross-modal retrieval without additional computational overhead (detailed in Section 3.3).

To train UEM and align the three modalities within a shared latent space, we introduce the *Dual-Sigmoid Alignment (DSA)* loss, a pairwise formulation based on the sigmoid contrastive objective. Given a batch of $b$ triplets $\{(q_i, I_i, T_i)\}_{i=1}^b$, the text-query loss is:

$$\mathcal{L}^T = -\frac{1}{b} \sum_{i=1}^b \sum_{j=1}^b \log \frac{1}{1 + \exp\{\gamma_{ij}(-\tau z_{ij}^T + \eta)\}}, \tag{2}$$

where $z_{ij}^T = \langle \mathbf{q}_i, \mathbf{T}_j \rangle$ is the inner product between the $i^{\text{th}}$ query and $j^{\text{th}}$ text embedding, $\gamma_{ij} \in \{+1, -1\}$ indicates whether the pair is positive or negative, and $\tau, \eta$ are learnable temperature and bias. An analogous loss $\mathcal{L}^I$ is defined for query–image pairs with $z_{ij}^I = \langle \mathbf{q}_i, \mathbf{I}_j \rangle$. Importantly, we update only $\mathcal{E}^T$ during training, keeping $\mathcal{E}^q$ and $\mathcal{E}^I$ frozen to preserve their pretrained multimodal alignment. A symmetric contrastive regularization further encourages $\mathcal{E}^T$ to stay consistent with the frozen encoders, yielding a unified embedding space. The final objective is:

$$\mathcal{L} = \lambda \mathcal{L}^T + (1 - \lambda) \mathcal{L}^I, \tag{3}$$

where $\lambda \in [0, 1]$ controls the relative weight of text and image alignment. Full training details are reported in Appendix D.

**Unified co-modality–informed retrieval (UCMR).** The core of our framework lies in the retrieval mechanism that effectively leverages the co-modality representations. As shown in Fig. 2(b), given a user query $q$, we first encode it into a dense vector representation $\mathbf{q}$. Offline, the visual and textual components of each document page $p_i$ are independently encoded into vectors $\mathbf{I}_i$ and $\mathbf{T}_i$ ($\forall i \in \{1, 2, ..., M\}$), respectively. The relevance of a page to the query is determined by calculating the inner product between the query vector and each modality's representation: $z_i^I = \langle \mathbf{q}, \mathbf{I}_i \rangle$ for the visual similarity and $z_i^T = \langle \mathbf{q}, \mathbf{T}_i \rangle$ for the textual similarity. A straightforward method to calculate the final score $s_i$ is to combine these two scores through a weighted linear combination:

$$s_i = \alpha z_i^T + (1 - \alpha) z_i^I, \quad \forall i \in \{1, 2, ..., M\}, \tag{4}$$

where $\alpha \in [0, 1]$ is a weighting parameter that balances the contribution of the textual modality. This linear combination may yield sub-optimal results due to the inherent differences in modality nature and encoder capabilities. Also, the scales and distributions of the raw inner products are not directly comparable. To address this, we propose a unified co-modality–informed retrieval method that normalizes the scores into a shared, comparable space.

As shown in Fig. 2(c), to shift the inner product scores of visual and textual modalities to a comparable space, we first apply a sigmoid function to normalize the inner product scores to a [0,1] range:

$$\bar{z}_i^I = \frac{1}{1 + \exp\{-z_i^I\}}, \quad \bar{z}_i^T = \frac{1}{1 + \exp\{-z_i^T\}}. \tag{5}$$

Empirically, we observed that the resulting distributions of $\bar{z}_i^I$ and $\bar{z}_i^T$ across a corpus approximate a Gaussian distribution. Therefore, to mitigate the bias from differing distribution parameters, we apply a Z-score normalization:

$$\tilde{z}_i^I = \frac{\bar{z}_i^I - \mu^I}{\sigma^I}, \quad \tilde{z}_i^T = \frac{\bar{z}_i^T - \mu^T}{\sigma^T}, \tag{6}$$

where $\mu^I = \frac{1}{M} \sum_{i=1}^M \bar{z}_i^I$ and $\sigma^I = \sqrt{\frac{1}{M} \sum_{i=1}^M (\bar{z}_i^I - \mu^I)^2}$ are the mean and standard deviation of the visual similarity scores, respectively (with $\mu^T$ and $\sigma^T$ defined analogously for the text modality). This step ensures the scores from both modalities are standardized to a common scale with zero mean and unit variance. The final, unified retrieval score $\tilde{s}_i$ is then calculated as:

$$\tilde{s}_i = \beta \tilde{z}_i^T + (1 - \beta) \tilde{z}_i^I, \quad \forall i \in \{1, 2, ..., M\}, \tag{7}$$

where $\beta$ represents the calibrated contribution of textual information. By eliminating the effects of distributional differences, the parameter $\beta$ more accurately denotes the relative confidence we place in the textual modality versus the visual modality for a given task.

### 3.3 COMPUTATIONAL COST ANALYSIS

A critical requirement for a practical RAG system is low-latency retrieval. We emphasize that the CMRAG framework introduces negligible extra latency during the online retrieval phase, making it highly feasible for real-time applications. In a deployed system, all visual documents are parsed offline. In addition, all document images and texts are encoded offline by $\mathcal{E}^I$ and $\mathcal{E}^T$, respectively, and their embeddings are precomputed and stored in the retrieval index. At query time, the user's query is encoded only once by $\mathcal{E}^q$. While the number of subsequent similarity calculations (inner products) is doubled compared to a single-modality retriever—as the query embedding is compared to both the text and image index—this operation is highly efficient on modern GPUs due to the parallelizable nature of matrix computations. Therefore, the online cost is dominated by a single encoding step, with the additional similarity calculations adding minimal overhead. This analysis confirms that CMRAG can achieve improved retrieval performance without a corresponding increase in computational cost.

## 4 DATA CONSTRUCTION

To effectively train and evaluate our CMRAG pipeline on visual documents, we construct both training and evaluation datasets. Overall, the training data is constructed based on existing visual

| | **MMLongBench** | **REAL-MM-RAG** | | | | **LongDocURL** |
|---|---|---|---|---|---|---|
| | Doc | FinReport | FinSlides | TechReport | TechSlides | Filtered |
| #Queries | 1082 | 853 | 1052 | 1294 | 1354 | 770 |
| #Documents | 135 | 19 | 65 | 17 | 61 | 134 |
| #Images | 6529 | 2687 | 2280 | 1674 | 1963 | 11034 |
| Avg. Query Length | 95.59 | 78.56 | 93.38 | 89.03 | 82.17 | 161.83 |
| Avg. Queries per Doc | 8.01 | 44.89 | 16.18 | 76.12 | 22.20 | 5.75 |
| Avg. Images per Doc | 48.36 | 141.42 | 35.08 | 98.47 | 32.18 | 82.34 |

Table 1: Statistics of evaluation datasets. We apply a unified parsing procedure to all datasets, and sample one-third of LongDocURL for testing to balance controllability and cost.

document RAG tasks, while the evaluation data covers mainstream benchmarks in visual document question answering. We will release all our processed datasets, including both training and evaluation sets, to facilitate further research by the community.

**Training data.** Our training corpus is built upon the synthetic dataset from VisRAG (Yu et al., 2025), which comprises diverse PDF documents (e.g., textbooks, academic papers, manuals) and approximately 240k query-document pairs generated by GPT-4o. To fully exploit the multimodal nature of these documents, we re-process all document pages (around 40k) using Qwen2.5-VL-7B (Bai et al., 2025) for end-to-end document parsing. This step extracts a structured representation for each page, including the entire page image, segmented sub-figures, and OCR-based text stored in HTML format. This enriched dataset provides fine-grained, aligned multimodal supervision, which is crucial for training our model to perform effective cross-modal retrieval and reasoning. Detailed data sources and processing prompts are provided in Appendix E.

**Evaluation data.** To comprehensively evaluate our method, we adopt several widely used VDQA benchmarks that span diverse domains and task types. Specifically, we include **MMLongBench** (Ma et al., 2024b)[1], **LongDocURL** (Deng et al., 2024)[2], and **REAL-MM-RAG** (Wasserman et al., 2025)[3]. These datasets cover industrial documents, scientific papers, presentation slides, and long multi-page documents, providing a broad and challenging testbed for assessing the retrieval and generation capabilities of our model. Detailed statistics of the evaluation datasets are provided in Tab. 1.

## 5 EXPERIMENTS

### 5.1 EXPERIMENT SETUP

**Baselines.** We evaluate the effectiveness of our proposed method against several strong embedding model baselines: (1) **BGE:** A state-of-the-art text embedding model that uses only the parsed textual content from the documents (Xiao et al., 2023); (2) **CLIP-B/32:** The CLIP model with a Vision Transformer-B/32 image encoder; (3) **CLIP-L/14-336:** A larger and higher-resolution CLIP variant with a Vision Transformer-L/14 image encoder pretrained on $336 \times 336$ pixel images (Radford et al., 2021); (4) **SigLIP:** The SigLIP model, which forms the foundation of our architecture (Zhai et al., 2023); and (5) **SigLIP2:** The latest iteration of the SigLIP model (Tschannen et al., 2025).

**Evaluation metrics.** We evaluate our method from two perspectives: (1) **Retrieval metrics.** We adopt standard measures including *Recall*, *nDCG*, and *MRR*. Detailed results for each dataset and metric are reported in the Appendix F. (2) **Generation metrics.** Traditional evaluation measures such as Exact Match (EM) and token-level F1 often fail to capture the semantic equivalence of multi-span or diverse gold answers, thereby underestimating model performance. To overcome this limitation, we employ a large language model (LLM) as an automatic judge to assess whether a generated response is correct, providing a fairer and more reliable comparison across different models.

---

[1] https://huggingface.co/datasets/MMlongBench/MMLongBench-Doc
[2] https://huggingface.co/datasets/THUDM/LongDocURL
[3] https://huggingface.co/collections/ibm-research

| Method | MMLongBench | REAL–MM–RAG | | | | LongDocURL |
|--------|-------------|-------------|--|--|--|------------|
| | Doc | Finreport | Finslides | Techreport | Techslides | Filtered |
| BGE (T) | 23.29% | **49.62%** | 32.55% | 42.44% | 65.01% | 11.69% |
| CLIP-B/32 (I) | 28.61% | 6.45% | 20.68% | 4.59% | 43.73% | 21.24% |
| CLIP-L/14-336 (I) | 41.17% | 37.61% | 58.08% | 36.80% | 63.25% | 50.27% |
| SigLIP (I) | 47.32% | 39.29% | 62.45% | 45.37% | 71.88% | 57.74% |
| SigLIP2 (I) | 42.02% | 1.45% | 4.93% | 3.03% | 6.78% | 5.43% |
| CMRAG-R [ours] (I+T) | **47.64%** | 41.85% | **67.97%** | **47.22%** | **78.10%** | **58.30%** |

Table 2: Retrieval results across six VDQA datasets with the evaluation metric of MRR@10. **Bold** and underlined values represent the best and second-best scores, respectively. We also report recall and nDCG in Appendix F.

| Method | MMLongBench | REAL–MM–RAG | | | | LongDocURL |
|--------|-------------|-------------|--|--|--|------------|
| | Doc | Finreport | Finslides | Techreport | Techslides | Filtered |
| CLIP-L/14-336 | 30.68% | 27.90% | 54.94% | 35.47% | 57.09% | 44.42% |
| SigLIP | **31.33%** | 29.54% | 55.99% | 41.65% | 62.26% | 47.66% |
| CMRAG [ours] | 31.05% | **32.13%** | **60.46%** | **45.60%** | **64.18%** | **48.18%** |
| Oracle (I) | 41.13% | 53.11% | **74.52%** | 71.72% | 75.48% | 64.81% |
| Oracle (I + T) | **43.25%** | **55.45%** | 70.72% | **73.42%** | **76.14%** | **67.79%** |

Table 3: Generation results across six VDQA datasets with top-3 retrievals. **Bold** values represent the best scores.

**Implementation details.** We implemented our CMRAG framework using the Qwen2.5 series of models. The backbone Vision Language Model (VLM) for both document parsing and final answer generation is **Qwen2.5-VL-7B-Instruct** (Bai et al., 2025). For evaluation, we employ **Qwen2.5-7B-Instruct** (Yang et al., 2025a) as the judge model. To ensure deterministic and reproducible outputs, we set the decoding temperature to 0.0 for all models. Our unified encoding model produces embeddings with a dimensionality of 1152, and the maximum length of the text encoder is 768. For retrieval, the top-$k$ value is set to 3. All models are implemented using the PyTorch framework and trained on NVIDIA H100 GPUs.

## 5.2 MAIN RESULTS

**Retrieval results.** The overall performance of our proposed **CMRAG-R** compared to all baselines is summarized in Tab. 2. Our model consistently outperforms all baseline methods on almost all benchmarks, demonstrating the effectiveness of our co-modality approach across diverse benchmarks. Interestingly, on the Finreport subset, the text-only BGE baseline performs the best among all baselines, indicating that this subset is highly text-dominant, where textual content is the primary carrier of information. Furthermore, the pattern that performance on Slides is generally higher than on Reports across most models highlights a key challenge: it is **inefficient** to process text-heavy visual documents (like reports) as pure images, as visual encoders struggle to capture dense textual information. It is important to note that our model is trained on a relatively small-scale and domain-limited dataset. Despite this, it achieves relatively strong performance, and as we explore in Section 5.3, there is significant potential for further gains by incorporating a larger training dataset, indicating a promising direction for future work. It is worth noting that *SigLIP2 performs significantly worse than SigLIP* in our experiments, mainly due to its design focus on dense prediction and multilingual capabilities, which makes it less stable for document retrieval tasks (see Appendix F for detailed analysis).

**Generation results.** As shown in Tab. 3, our **CMRAG** framework consistently outperforms all baseline retrieval methods, demonstrating that high-quality co-modality retrieval is crucial for VDQA tasks. This conclusion is further validated by the Oracle experiments, which show that providing the generator with ground-truth evidence from both image (I) and text (T) modalities yields the highest accuracy, confirming the complementary value of both information sources. On MMLong-Bench, our CMRAG score is slightly lower than the SigLIP generation baseline. This is potentially

| Method | MMLongBench | Finreport | Finslides | Techreport | Techslides | LongDocURL |
|--------|-------------|-----------|-----------|------------|------------|------------|
| CMRAG-R | **47.64%** | **41.85%** | **67.97%** | **47.22%** | **78.10%** | **58.30%** |
| w/o norm | 44.46% | 29.67% | 60.61% | 31.12% | 74.94% | 53.43% |

Table 4: Ablation study on unified co-modality retrieval. **Bold** values represent the best scores.

| Method | MMLongBench | Finreport | Finslides | Techreport | Techslides | LongDocURL |
|--------|-------------|-----------|-----------|------------|------------|------------|
| BGE (T) | 23.29% | 49.62% | 32.55% | 42.44% | 65.01% | 11.69% |
| SigLIP (I) | 47.32% | 39.29% | 62.45% | 45.37% | 71.88% | 57.74% |
| SigLIP + BGE (I+T) | **47.48%** | **57.49%** | **67.97%** | **54.94%** | **79.44%** | **64.90%** |

Table 5: Ablation study on multi-mode embedding model capability.

because the dataset contains *20.8% of questions that are not answerable*; providing rich textual context may prompt the generator to attempt an answer, leading to errors. Furthermore, the observation that abundant input can sometimes be detrimental is exemplified in the Oracle results for Finslides, where using only images (I) leads to better performance than using both images and text (I+T). Therefore, **dynamically controlling the modality and quantity of retrieved context** presents a promising future direction to enhance both the efficiency and accuracy of multimodal RAG systems.

## 5.3 ABLATION STUDY

**Co-modality similarity scores should be unified.** Tab. 4 presents the results of an ablation study validating the necessity of unifying similarity scores across modalities. The row **w/o norm** shows the performance when the image-query and text-query similarity scores are not normalized before fusion. Compared to our CMRAG-R, which uses a unified scoring approach, the performance drops significantly across all benchmarks, demonstrating that proper normalization is crucial for effectively combining co-modality signals.

**The unified encoding model can be further enhanced with larger datasets.** An analysis of a strong ensemble baseline, **SigLIP + BGE**, provides a promising direction for future work. This baseline uses SigLIP to encode images and the query, and employs BGE—a text embedding model trained on a significantly larger dataset—to encode the parsed text and query (**encoding queries twice**). As shown in Tab. 5, this ensemble achieves superior performance, particularly on text-heavy report-style documents. This result demonstrates that the retrieval performance of our proposed unified encoding model (UEM) is not yet saturated and can be substantially enhanced in the future by scaling up the training data. More results can be found in Appendix H.

## 5.4 ANALYSIS

**Weight of text-query modality.** In our framework, the entire page image serves as the primary carrier of information, as it inherently contains all the visual and textual content. Therefore, the image-query similarity is assigned a dominant weight. The text-query similarity acts as a compensatory signal to capture fine-grained semantic details that may be lost in the image representation. In this study, we set the text modality weight to $\beta = 0.1$, reflecting this design principle where the image modality is dominant and the parsed text provides a complementary semantic boost.

**Unified distribution of similarity scores.** Fig. 3 illustrates the unified distributions of query-image (Sim-I) and query-text (Sim-T) similarity scores after applying our normalization method across the (a) Finslides, (b) Techslides, and (c) LongDocURL benchmarks. The resulting distributions are closely aligned, with low Kullback–Leibler divergence ($D_{\mathrm{KL}}$) values of 0.132, 0.049, and 0.094, respectively. This high degree of distributional similarity indicates that our proposed statistical normalization method effectively mitigates the inherent discrepancies between modalities, providing a reasonable and stable foundation for co-modality score fusion.

**Case study.** As illustrated in Fig. 4, a case study from *finreport* further demonstrates the practical effectiveness of our CMRAG framework. For a given query, the ground truth (GT) evidence is located on page 33. Our CMRAG retriever correctly identifies this GT page as the most similar

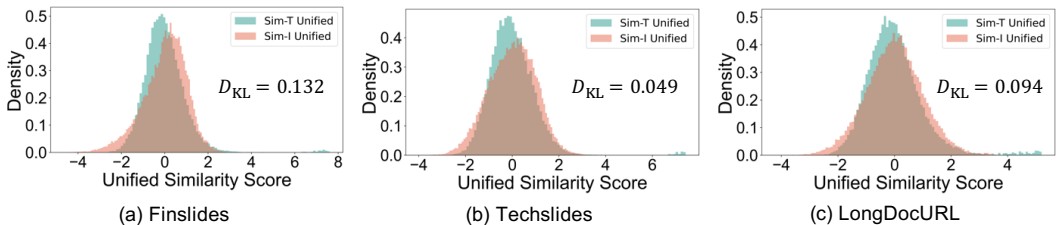

Figure 3: Unified distributions of query-image (Sim-I) and query-text (Sim-T) similarity scores of (a) Finslides, (b) Tehslides, and (c) LongDocURL.

result. On the contrary, the SigLIP baseline, which relies solely on the image modality, ranks the GT page only as the second most similar. Furthermore, even when the GT page is provided directly to the generator, the baseline fails to produce an accurate answer, whereas our method succeeds. This suggests that for pages with dense textual content, treating them as pure images is insufficient for accurate comprehension. The generator requires the explicit textual information that our method provides. This case underscores the dual advantage of CMRAG: it improves both retrieval accuracy and the quality of the final generated answer. We provide more cases in Appendix G.

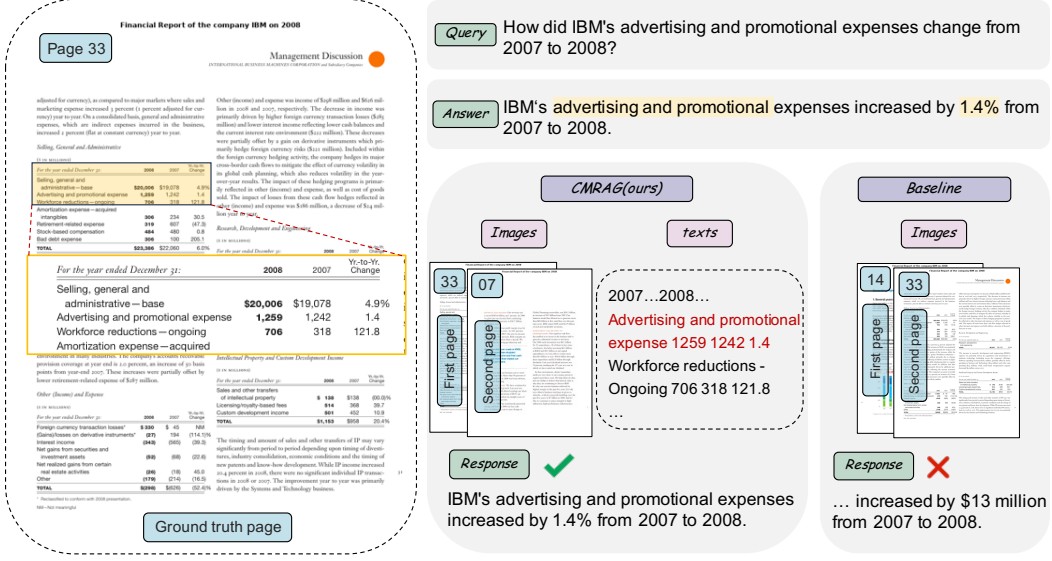

Figure 4: A case comes from Finreport.

## 6    CONCLUSION

We introduced CMRAG, a novel co-modality RAG framework that overcomes the limitations of text-only and image-only approaches by unifying textual and visual information for retrieval and generation on visual documents. Central to our method is a unified encoding model that projects queries, images, and parsed texts into a shared latent space, enabling effective co-modality retrieval. Extensive experiments demonstrate that CMRAG consistently outperforms strong baselines, validating that leveraging both modalities is superior for visual document QA tasks. Our analysis also reveals important insights: text-dominant documents benefit from explicit textual retrieval, while overly abundant context can sometimes hinder performance on unanswerable questions, pointing to dynamic input control as a key direction for future work.

## ETHICS STATEMENT

This work aims to advance multimodal document retrieval and question answering. All data used in this study come from publicly available resources or standard benchmark datasets, and no private or sensitive information is involved. Some training queries were automatically generated by large language models based on open-source data, and are solely intended for academic research without containing any personal or sensitive content. We strictly adhere to the original licenses of all datasets, use the data only for non-commercial academic purposes, and release only processed representations or derived resources. We acknowledge that multimodal large models may inherit biases from training data, which could affect retrieval and generation results. Therefore, our method should not be directly applied to high-stakes domains such as healthcare, law, or finance, and should be carefully validated with necessary human oversight before deployment. We are also aware of the energy consumption associated with training and inference of large models. To mitigate environmental impact, we reuse existing pretrained models whenever possible and optimize computational efficiency during experiments. This work is intended solely for academic and industrial research and should not be applied to misinformation generation, surveillance, or other potentially harmful uses.

## REPRODUCIBILITY STATEMENT

We have made extensive efforts to ensure the reproducibility of our work. Detailed descriptions of the model architecture, training objectives, and computational cost analysis are provided in Section 3 and Appendix E.1. The construction and processing steps of both training and evaluation datasets are explained in Section 4 and Appendix E, with benchmark datasets referenced in Table 1. Prompt templates used for document parsing, generation, and evaluation are included in Appendix B. Experimental setups, hyperparameters, and implementation details are presented in Section 5.1. Moreover, we report comprehensive retrieval and generation results, ablations, and case studies in Section 5 and Appendix F to facilitate verification. We will release the processed datasets and codebase upon acceptance to further support reproducibility by the research community.

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

# A    NOTATION LIST

The main notations and abbreviations used in this paper are listed in Tab. 6.

| Notation | Explanation |
|---|---|
| $\mathcal{D}$ | Corpus of visual documents |
| $p_i$ | $i^{\text{th}}$ document page in the corpus |
| $M$ | Total number of pages in the corpus |
| $P_k$ | Retrieved top-$k$ document pages |
| $\mathcal{R}$ | Retriever |
| $\mathcal{P}$ | Prompt |
| $\mathcal{G}$ | VLM generator |
| $q$ | Query |
| $\mathbf{q}$ | Query embedding |
| $\hat{a}$ | Generated answer |
| $\mathcal{V}$ | VLM parser |
| $\mathcal{E}^q$ | Query encoder |
| $\mathcal{E}^I$ | Image encoder |
| $\mathcal{E}^T$ | Text encoder |
| $I_i$ | Visual presentation of $i^{\text{th}}$ document page |
| $T_i$ | Textual presentation of $i^{\text{th}}$ document page |
| $\mathbf{I}_i$ | Embedding of $I_i$ |
| $\mathbf{T}_i$ | Embedding of $T_i$ |
| $d$ | Embedding dimension |
| $z_i^I$ | Inner product of $\mathbf{q}$ and $\mathbf{I}_i$ |
| $z_i^T$ | Inner product of $\mathbf{q}$ and $\mathbf{T}_i$ |
| $\bar{z}_i^I$ | Normalized $z_i^I$ using sigmoid function |
| $\bar{z}_i^T$ | Normalized $z_i^T$ using sigmoid function |
| $\tilde{z}_i^I$ | Z-score Normalized $\bar{z}_i^I$ |
| $\tilde{z}_i^T$ | Z-score Normalized $\bar{z}_i^T$ |
| $\mu^I$ | Mean value of $\bar{z}_i^I$ |
| $\mu^T$ | Mean value of $\bar{z}_i^T$ |
| $\sigma^I$ | Standard deviation of $\bar{z}_i^I$ |
| $\sigma^T$ | Standard deviation of $\bar{z}_i^T$ |
| $s_i$ | Weighted similarity score |
| $\tilde{s}_i$ | Weighted normalized similarity score |
| $\alpha\ (\beta)$ | Weight for the textual modality in $s_i$ ($\tilde{s}_i$) |
| $b$ | Batch size during training |
| $\tau/\eta$ | Temperature/bias for stable training |
| $\gamma_{ij}$ | Pair indicator |
| $\mathcal{L}^T$ | Sigmoid loss for text-query modality |
| $\mathcal{L}^I$ | Sigmoid loss for iamge-query modality |
| $\mathcal{L}$ | Final loss for training |
| $\lambda$ | Weight for $\mathcal{L}^T$ in $\mathcal{L}$ |

| Abbreviation | Explanation |
|---|---|
| VLM | Vision-language model |
| CMRAG | Co-modality–based RAG |
| UEM | Unified encoding model |
| DSA | Dual-Sigmoid alignemnt |
| UCMR | Unified co-modality–informed retrieval |
| MMRAG | Multi-modality RAG |
| VDQA | visual document question-answering |

Table 6: Explanation of main notations and abbreviations used in this study.

# B   PROMPT TEMPLATE

---

**(a) *Prompt for parsing images***

You are an AI specialized in recognizing and extracting text from images. Your mission is to analyze the image document and generate the result in QwenVL Document Parser HTML format using specified tags while maintaining user privacy and data integrity.

Image: {image_path}

---

**(b) *Prompt for generating answers based on entire images***

Your task is to answer the question based on the provided images. Please directly provide the answer inside <answer> and </answer>, without detailed illustrations. For example, <answer> Beijing </answer>.

Image: {image_path(s)}

Question: {query}

---

**(c) *Prompt for generating answers based on sub-images and text***

Your task is to answer the question based on the provided sub-images and their parsed text. Please directly provide the answer inside <answer> and </answer>, without detailed illustrations. For example, <answer> Beijing </answer>.

Image: {image_path(s)}

Parsed Text: {parsed_text}

Question: {query}

---

**(d) *Prompt for generating answers based on entire images and text***

Your task is to answer the question based on the provided images and their parsed text. Please directly provide the answer inside <answer> and </answer>, without detailed illustrations. For example, <answer> Beijing </answer>.

Image: {image_path(s)}

Parsed Text: {parsed_text}

Question: {query}

---

**(e) *Prompt for judging generated answers***

You are an expert evaluation system for a question answering chatbot.

You are given the following information:

- the query

- a generated answer

- a reference answer

Your task is to evaluate the correctness of the generated answer.

**Query**

{q}

**Reference Answer**

{gold_ans}

**Generated Answer**

{gen_ans}

Your response should be formatted as following:

<judge>True or False</judge>

If the generated answer is correct, please set "judge" to True. Otherwise, please set "judge" to False.

Please note that the generated answer may contain additional information beyond the reference answer.

---

Figure 5: Prompt templates for (a) parsing images, (b) generating answers based on entire images, (c) generating answers based on sub-images and text, generating answers based on entire images and text, and (e) judging generated answers. The first template can be found at `https://github.com/QwenLM/Qwen2.5VL/blob/main/cookbooks/document_parsing.ipynb` and the rest can be referred to (Wang et al., 2025b).

## C ADDITIONAL RELATED WORK

**Video–based MMRAG** refers to retrieving videos from the corpus to help answer given queries (Caba Heilbron et al., 2015; Xu et al., 2016; Anne Hendricks et al., 2017; Wang et al., 2019; Kriz et al., 2025; Wan et al., 2025). Since encoding videos may incur high computational costs, a few studies pre-processed videos using VLMs and converted videos to textual modality (Zhang et al., 2024a; Arefeen et al., 2024; Ma et al., 2025). For example, (Zhang et al., 2024a) first detected key information in videos such as human faces, based on which a VLM was prompted to generate scene captions for video frames. Consequently, the video modality can be converted to a text modality, which can significantly reduce computational costs and facilitate retrieval and generation. Furthermore, a few studies (Luo et al., 2024; Jeong et al., 2025; Reddy et al., 2025) processed videos by selecting or clustering representative frames, facilitating video retrieval and final generation.

## D TRAINING DETAILS

The balancing hyperparameter $\lambda$ in the training objective is set to 0.5, giving equal weight to the text and image modality losses. The temperature and bias parameters in the sigmoid loss are initialized to 10 and -10, respectively. We train the model for 32 epochs with a batch size of 32 distributed across 8 NVIDIA H100 GPUs, using the AdamW optimizer with a learning rate of $3 \times 10^{-5}$ and a weight decay of 0.05. The embedding dimension is 1152, with a maximum sequence length of 768 for the text encoder. The number of trainable parameters is approximately 0.45B, while the frozen parameters amount to 0.88B. For online deployment, only the 0.4B parameter query encoder is required to process user queries.

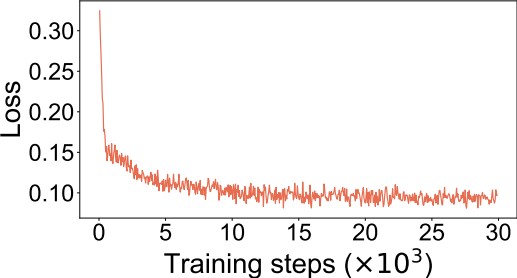

Figure 6: training process.

Fig. 6 illustrates the training process. The loss decreases rapidly within the first 1k steps, indicating quick initial learning. The rate of decrease slows considerably after 5k steps, and the loss stabilizes, showing convergence after approximately 15k steps. This progression confirms that the model successfully learns from the training dataset.

## E DATA CONSTRUCTION DETAILS

### E.1 TRAINING DATA SOURCE AND COMPOSITION

Our training corpus is based on the training data provided by VisRAG, constructed from crawled multimodal documents with synthetic query–document pairs automatically generated by prompting GPT-4o.

**Synthetic data.** We collected documents from several publicly available sources, including OpenStax (college-level textbooks), ICML/NeurIPS 2023 (academic papers), and ManualsLib (product manuals). GPT-4o was then prompted to generate queries for these documents, resulting in approximately 239k synthetic query–document pairs. This large-scale corpus covers both text-dominated and image-dominated content, enabling the model to learn retrieval and reasoning over complex multimodal structures.

**Summary**   Tab. 7 summarizes the training datasets used in this work, including their sources, document types, page counts, and the number of query–document pairs.

Table 7: Training datasets used in our work. Synthetic data are constructed by prompting GPT-4o to generate queries on crawled documents, while VQA datasets are filtered to retain retrieval-suitable queries.

| Dataset | Source | Type | # Pages | Train # Q-D Pairs |
|---|---|---|---|---|
| Textbooks | OpenStax | College-level textbooks | 10,000 | - |
| ICML Papers | ICML 2023 | Academic papers | 5,000 | - |
| NeurIPS Papers | NeurIPS 2023 | Academic papers | 5,000 | - |
| Manuallib | ManualsLib | Product manuals | 20,000 | - |
| **Synthetic** | **Various (above)** | **Crawled + GPT-4o queries** | - | 239,358 |

## E.2   RE-PROCESSING DETAILS

To further enhance the usability of our training corpus, we re-processed all documents using the Qwen2.5-VL-7B model, which is capable of end-to-end document parsing. Instead of relying solely on OCR tools or manually designed pipelines, the model directly converts page images into structured representations in a unified HTML format. Each page is parsed into three key components: (1) the **entire page image**, (2) **sub-image regions** such as tables, figures, and diagrams, and (3) **OCR-based structured text**, which is saved in an HTML file.

The parsing was guided by a carefully designed system prompt to ensure both structured formatting and data integrity. The actual prompt is listed in Appendix B.

This re-processing step provides explicit, structured multimodal representations, bridging the gap between purely text-based and purely image-based datasets. As a result, it allows the model to capture fine-grained semantic information and improves the alignment between visual and textual modalities, which is essential for cross-modal retrieval and reasoning tasks.

## F   RETRIEVAL EXPERIMENT RESULTS

Tab. 8 reports retrieval results across six VDQA datasets. We observe several key findings: (1) Single-modality baselines perform unevenly—BGE (text-only) excels on text-heavy domains such as Finreport and Techreport, while CLIP-based models (image-only) perform better on visually structured datasets like Finslides and Techslides. (2) SigLIP achieves the best balance among single-modality methods, significantly outperforming CLIP variants, while SigLIP2 underperforms across all datasets, confirming that stronger vision–language pretraining does not necessarily transfer to document retrieval. (3) The ensemble baseline SigLIP + BGE further boosts performance, indicating the complementary of textual and visual signals. (4) Our proposed method CMRAG-R consistently matches or surpasses SigLIP and achieves competitive performance against SigLIP+BGE, particularly on Finslides, Techreport, and LongDocURL. These results highlight the effectiveness of co-modality retrieval in leveraging both text and image information without requiring external ensembles.

As discussed in Section 5, we observed that SigLIP2 performs significantly worse than SigLIP across multiple benchmarks. We attribute this degradation to several factors: (1) its pretraining objective emphasizes dense prediction and multilingual understanding, which introduces a more complex embedding space less suited for pure retrieval ranking; (2) its architecture and tokenizer are less optimized for long-form document text, while our benchmarks are text-heavy; and (3) domain mismatch, as SigLIP2 is mainly trained on web-scale multilingual captioning data, which diverges from structured documents such as financial reports or scientific papers. We verified our implementation against official checkpoints and reproduced this behavior across multiple runs, confirming that the performance drop is inherent to the model rather than due to experimental errors. This finding suggests that stronger vision–language pretraining does not always transfer to document retrieval and further motivates the need for tailored co-modality approaches such as ours.

| Model | Metric | MMLongBench | Finreport | Finslides | Techreport | Techslides | LongDocURL |
|---|---|---|---|---|---|---|---|
| BGE | Recall@1 | 8.45% | 38.45% | 21.96% | 32.92% | 55.02% | 2.53% |
| | Recall@5 | 22.84% | 64.95% | 46.01% | 55.41% | 77.62% | 8.97% |
| | Recall@10 | 34.50% | 73.97% | 62.45% | 65.69% | 85.38% | 15.25% |
| | nDCG@1 | 13.31% | 38.45% | 21.96% | 32.92% | 55.02% | 6.23% |
| | nDCG@5 | 18.51% | 52.56% | 34.23% | 44.63% | 67.37% | 7.35% |
| | nDCG@10 | 22.96% | 55.46% | 39.55% | 47.95% | 69.92% | 9.92% |
| | MRR@1 | 13.31% | 38.45% | 21.96% | 32.92% | 55.02% | 6.23% |
| | MRR@5 | 21.19% | 48.43% | 30.36% | 41.07% | 63.93% | 10.15% |
| | MRR@10 | 23.29% | 49.62% | 32.55% | 42.44% | 65.01% | 11.69% |
| CLIP-B/32 | Recall@1 | 12.31% | 3.17% | 11.88% | 1.47% | 31.09% | 7.81% |
| | Recall@5 | 33.08% | 10.20% | 31.65% | 8.50% | 61.08% | 23.17% |
| | Recall@10 | 47.89% | 18.99% | 49.71% | 16.69% | 75.85% | 34.18% |
| | nDCG@1 | 17.93% | 3.17% | 11.88% | 1.47% | 31.09% | 11.82% |
| | nDCG@5 | 25.74% | 6.50% | 21.56% | 4.74% | 46.57% | 17.92% |
| | nDCG@10 | 30.98% | 9.31% | 27.39% | 7.35% | 51.32% | 22.04% |
| | MRR@1 | 17.93% | 3.17% | 11.88% | 1.47% | 31.09% | 11.82% |
| | MRR@5 | 26.60% | 5.31% | 18.28% | 3.54% | 41.79% | 19.55% |
| | MRR@10 | 28.61% | 6.45% | 20.68% | 4.59% | 43.73% | 21.24% |
| CLIP-L/14-336 | Recall@1 | 23.87% | 26.26% | 43.16% | 26.12% | 50.89% | 25.81% |
| | Recall@5 | 46.01% | 52.17% | 77.19% | 50.46% | 79.39% | 53.74% |
| | Recall@10 | 57.42% | 65.77% | 89.26% | 62.98% | 88.85% | 66.47% |
| | nDCG@1 | 31.15% | 26.26% | 43.16% | 26.12% | 50.89% | 37.53% |
| | nDCG@5 | 39.07% | 39.85% | 61.64% | 38.98% | 66.35% | 45.26% |
| | nDCG@10 | 43.26% | 44.27% | 65.58% | 42.99% | 69.42% | 50.08% |
| | MRR@1 | 31.15% | 26.26% | 43.16% | 26.12% | 50.89% | 37.53% |
| | MRR@5 | 39.83% | 35.77% | 56.44% | 35.17% | 61.97% | 48.66% |
| | MRR@10 | 41.17% | 37.61% | 58.08% | 36.80% | 63.25% | 50.27% |
| SigLIP | Recall@1 | 28.31% | 27.43% | 49.05% | 33.62% | 61.60% | 31.13% |
| | Recall@5 | 53.33% | 55.10% | 80.13% | 61.67% | 86.26% | 62.34% |
| | Recall@10 | 64.12% | 67.76% | 90.68% | 73.26% | 92.84% | 73.78% |
| | nDCG@1 | 37.43% | 27.43% | 49.05% | 33.62% | 61.60% | 44.55% |
| | nDCG@5 | 45.78% | 41.93% | 65.82% | 48.26% | 74.82% | 53.25% |
| | nDCG@10 | 49.67% | 46.06% | 69.23% | 52.01% | 76.95% | 57.62% |
| | MRR@1 | 37.43% | 27.43% | 49.05% | 33.62% | 61.60% | 44.55% |
| | MRR@5 | 46.17% | 37.58% | 61.04% | 43.82% | 71.00% | 56.52% |
| | MRR@10 | 47.32% | 39.29% | 62.45% | 45.37% | 71.88% | 57.74% |
| SigLIP2 | Recall@1 | 23.46% | 0.23% | 1.33% | 1.16% | 1.55% | 1.37% |
| | Recall@5 | 47.32% | 2.11% | 8.46% | 4.87% | 11.60% | 4.99% |
| | Recall@10 | 59.54% | 6.45% | 22.15% | 10.05% | 28.43% | 11.22% |
| | nDCG@1 | 31.98% | 0.23% | 1.33% | 1.16% | 1.55% | 2.60% |
| | nDCG@5 | 40.01% | 1.22% | 4.45% | 2.97% | 6.30% | 3.84% |
| | nDCG@10 | 44.39% | 2.57% | 8.80% | 4.63% | 11.68% | 6.07% |
| | MRR@1 | 31.98% | 0.23% | 1.33% | 1.16% | 1.55% | 2.60% |
| | MRR@5 | 40.67% | 0.93% | 3.19% | 2.35% | 4.60% | 4.31% |
| | MRR@10 | 42.02% | 1.45% | 4.93% | 3.03% | 6.78% | 5.43% |
| SigLIP + BGE | Recall@1 | 28.31% | 46.54% | 54.56% | 42.97% | 69.94% | 36.50% |
| | Recall@5 | 53.83% | 72.45% | 86.60% | 71.72% | 93.13% | 65.80% |
| | Recall@10 | 64.12% | 83.00% | 92.78% | 81.61% | 97.05% | 77.20% |
| | nDCG@1 | 37.43% | 46.54% | 54.56% | 42.97% | 69.94% | 53.60% |
| | nDCG@5 | 45.99% | 60.21% | 72.04% | 57.99% | 82.48% | 58.40% |
| | nDCG@10 | 49.67% | 63.56% | 74.00% | 61.15% | 83.75% | 62.70% |
| | MRR@1 | 37.43% | 46.54% | 54.56% | 42.97% | 69.94% | 53.60% |
| | MRR@5 | 46.45% | 56.25% | 67.15% | 53.65% | 78.92% | 63.70% |
| | MRR@10 | 47.48% | 57.49% | 67.97% | 54.94% | 79.44% | 64.90% |
| CMRAG-R(ours) | Recall@1 | 28.95% | 30.48% | 54.56% | 34.62% | 67.65% | 31.30% |
| | Recall@5 | 53.50% | 58.03% | 86.60% | 64.30% | 91.95% | 62.60% |
| | Recall@10 | 64.12% | 70.81% | 93.25% | 76.28% | 97.49% | 74.50% |
| | nDCG@1 | 38.17% | 30.48% | 54.56% | 34.62% | 67.70% | 45.20% |
| | nDCG@5 | 49.75% | 44.29% | 72.04% | 50.11% | 81.00% | 61.10% |
| | nDCG@10 | 52.10% | 48.52% | 74.00% | 54.14% | 82.80% | 63.80% |
| | MRR@1 | 38.17% | 30.48% | 54.56% | 34.62% | 67.70% | 45.20% |
| | MRR@5 | 46.50% | 40.08% | 67.15% | 45.56% | 77.30% | 56.90% |
| | MRR@10 | 47.64% | 41.85% | 67.97% | 47.22% | 78.10% | 58.30% |

Table 8: Retrieval results across six VDQA datasets.

# G  ADDITIONAL CASE STUDY

The case study illustrated in Fig. 7 highlights the strengths and limitations of different Oracle settings in handling complex queries. First, when relying solely on entire images, the VLM misinterprets the numbers, producing an incorrect output of 36 instead of 62. This demonstrates that VLMs may struggle to accurately ground numeric reasoning based solely on visual inputs. Similarly, when only sub-images plus parsed text are used, the model fails to capture the complete context, yielding partial and incomplete answers. The problem arises because the VLM failed to accurately parse the sub-image but extracted textual numbers only. However, incorporating entire images together with parsed text enables the model to generate the correct multi-span answer, as the parsed text provides a reliable textual grounding that compensates for the VLM's difficulty in interpreting fine-grained visual details. This shows that parsed text can serve as an essential complement, ensuring accurate reasoning across multiple evidence spans.

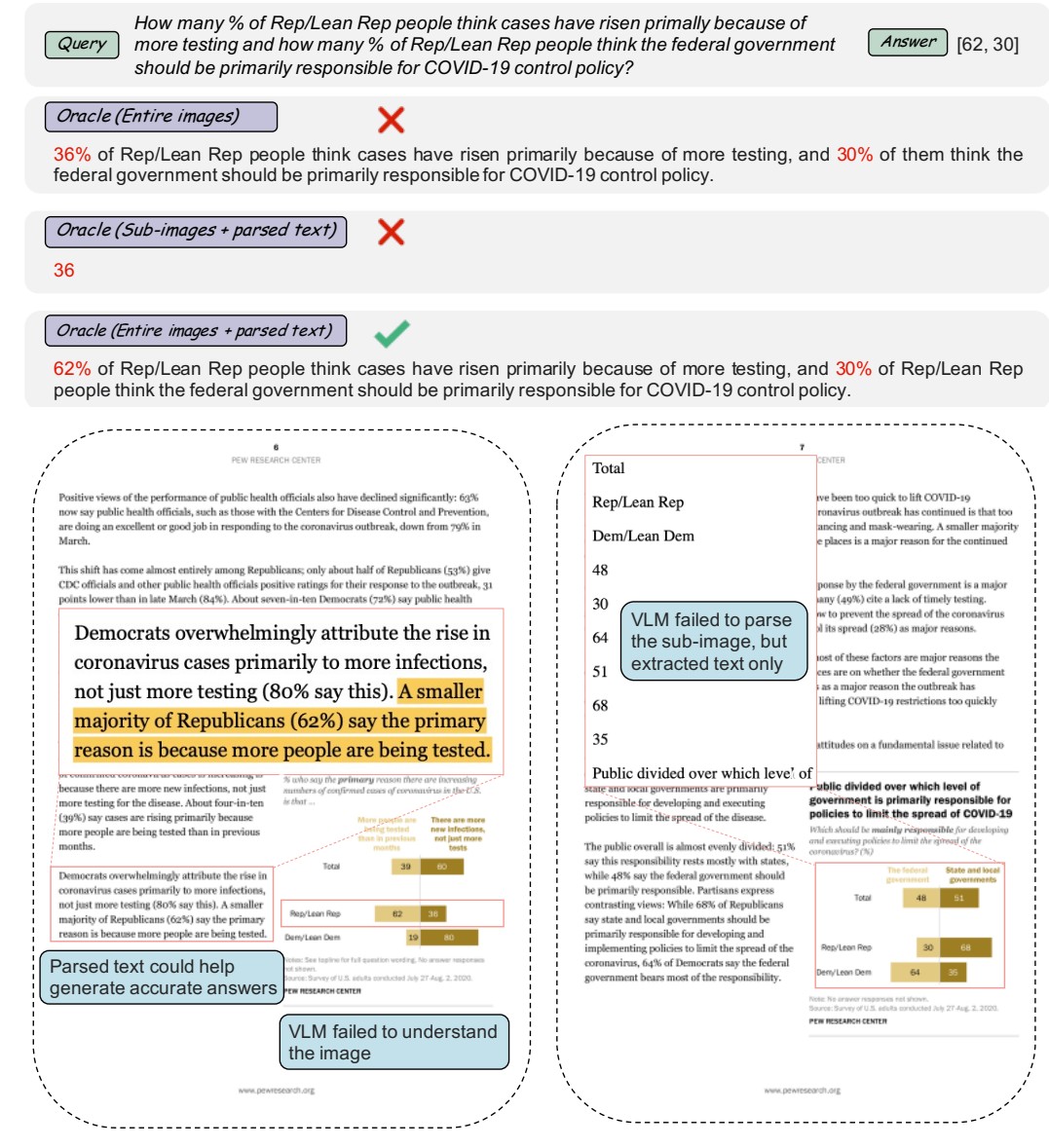

Figure 7: Qualitative comparison among three baselines.

## H    UNIFIED DISTRIBUTIONS OF DIFFERENT MODELS

Fig. 8 illustrates the unified distributions of similarity scores from the BGE and SigLIP models across the (a) Finslides, (b) Techslides, and (c) LongDocURL benchmarks after applying our normalization method. The resulting distributions are highly similar, as evidenced by the very low Kullback–Leibler divergence ($D_{\text{KL}}$) values of 0.010, 0.011, and 0.005, respectively. This demonstrates that our proposed normalization method is not only effective for our unified encoder but also generalizes well to different, independent models, providing a robust foundation for co-modality score fusion.

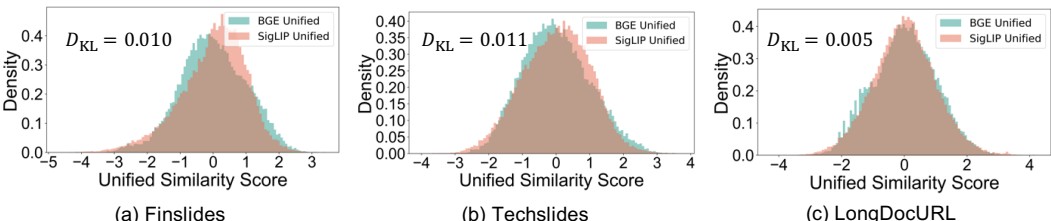

(a) Finslides            (b) Techslides            (c) LongDocURL

Figure 8: Unified distributions of BGE and SigLIP similarity scores of (a) Finslides, (b) Tehslides, and (c) LongDocURL.

## I    USE OF LLMS

In the preparation of this paper, large language models (LLMs) were used solely for the purpose of polishing the writing, including grammar correction, improving sentence fluency, and ensuring a consistent academic tone. All core intellectual content—including the conceptualization of the proposed method, the design and execution of experiments, the analysis and interpretation of results, and the conclusions drawn—is the original work of the authors. The authors take full responsibility for the entire content of this paper, including any text generated with the assistance of LLMs.

