# OpenReview forum: "CMRAG: Co-modality-based visual document retrieval and question answering"
_ICLR.cc/2026/Conference — ICLR 2026 Conference Withdrawn Submission_

### Official Review · Reviewer_btum · 2025-10-17

**Soundness:** 3
**Presentation:** 4
**Contribution:** 2
**Rating:** 4
**Confidence:** 5

**Summary:**

This paper presents a method for multi-page document retrieval. The approach encodes the query, document images, and document text content separately, then computes the similarity between the query and each page’s image as well as between the query and each page’s text. These similarities are fused through a weighted operation with regularized normalization, resulting in a final similarity ranking. The top three results are selected and fed into a multimodal large model for question answering.

The paper is clearly written and free of ambiguity; however, it lacks sufficient innovation and the comparative experiments are not comprehensive enough.

**Strengths:**

The paper is clearly written and easy to read.

**Weaknesses:**

The paper has the following shortcomings:

1. **Questionable distinction between online and offline processing.**
   The authors claim that only the query encoding is performed online, while all document-related processing is done offline. However, in practical applications, documents are often newly provided by users at runtime rather than being pre-stored and pre-encoded. In such common scenarios, the system would need to process the document, extract text, and encode representations online. Therefore, the paper’s description of computational efficiency based on an “offline” assumption seems somewhat unconvincing.

2. **Dependence on OCR.**
   The method still relies on an OCR stage. However, mainstream approaches in recent years have been evolving toward *OCR-free* pipelines. Introducing an OCR process (whether through VLMs or traditional OCR tools) inevitably increases processing time and may reduce overall efficiency.

3. **Limited novelty and incomplete innovation.**
   The core idea of the paper is simply to combine the similarity scores from query–image and query–text retrieval via direct weighting. Although normalization and regularization are applied in between, these are not truly innovative techniques. Essentially, this weighted-sum strategy represents a straightforward trade-off between textual and visual features in documents, controlled by a manually set hyperparameter β.
   However, a major challenge in the **DocVQA** field lies in effectively modeling both textual and *layout/visual* information. Layout features exist only in document images and are lost once OCR is applied. For example, questions such as “Is the chart located at the top or bottom of the page?” or “On which page does the red-highlighted text appear?” can only be answered by modules that process images directly, not by text-only modules. Yet the proposed system merely fuses text and image similarity scores without considering the nature of different question types. Consequently, for visually dependent questions, text-based similarity scores would be meaningless and introduce significant noise after normalization, potentially interfering with the visual similarity results. The weighting parameter β should therefore be **dynamic** and adaptive to question type, rather than fixed across all cases.

4. **Insufficient experiments.**
   Both the main and ablation experiments are too limited in scale. The main experiments only compare a few CLIP-based models — CLIP-B/32 (I), CLIP-L/14-336, SigLIP, and SigLIP — along with a classical text retrieval model (BGE). Yet several strong multimodal retrieval methods have emerged recently, such as **VDocRAG**, **ColPali**, **VISRAG**, and **Document Screenshot Embedding (DSE)**, none of which are included for comparison. Moreover, the generation module relies solely on **Qwen2.5-VL-7B-Instruct**, making the generation results less generalizable. The small experimental scope significantly reduces the value and credibility of the findings.

Thank you for your submission. I hope my comments are helpful for improving the paper.

**Questions:**

The paper is clearly written, and since I am familiar with this research area, I have no further questions.

---

### Official Review · Reviewer_DegG · 2025-10-26

**Soundness:** 3
**Presentation:** 3
**Contribution:** 2
**Rating:** 2
**Confidence:** 4

**Summary:**

The paper proposes CMRAG, a co-modality–based retrieval-augmented generation (RAG) framework designed for visual document question answering (VDQA).
Instead of relying solely on text (via OCR and layout parsing) or image modalities (via visual-language models), CMRAG unifies both text and image representations.
Its main components are:
Unified Encoding Model (UEM): jointly embeds queries, parsed text, and images into a shared latent space using triplet-based training with a dual-sigmoid alignment loss.
Unified Co-Modality-Informed Retrieval (UCMR): normalizes and fuses similarity scores from the two modalities via Z-score normalization, mitigating cross-modal distribution mismatch.
They also build a large-scale (query, text, image) triplet dataset and evaluate CMRAG on several VDQA benchmarks.
CMRAG outperforms single-modality RAG baselines (e.g., BGE, CLIP, SigLIP) in both retrieval and generation metrics while introducing minimal computational overhead.

**Strengths:**

1. UEM and UCMR are conceptually simple yet well-justified; they enable unified representation learning and normalized score fusion.
2. demonstrates negligible online latency, which is an important consideration for RAG systems.
3. releases a new large-scale triplet dataset to support future co-modality research.
4. explores effects of modality weighting, normalization, and dataset scale; provides qualitative case studies.

**Weaknesses:**

1. The proposed method is primarily an integration of existing components rather than a fundamentally new retrieval paradigm. The paper’s central idea (unifying textual and visual modalities for retrieval) is conceptually intuitive and has been explored in related works. The work lacks a distinct new principle beyond integrating existing best practices under a “co-modality” framing.
2. The claim that UEM produces a “shared embedding space” is not fully supported by analysis. There’s no evaluation of cross-modal alignment quality (e.g., retrieval between modalities or zero-shot transfer) to validate that the embeddings truly capture a unified semantic manifold.
3. There is no exploration of learnable or query-adaptive weighting mechanisms (e.g., uncertainty-based fusion, gating networks). The optimal balance likely varies by domain, but the model ignores this heterogeneity.
4. Although improvements are consistent, they are often small. The paper does not report variance across seeds or confidence intervals, which weakens claims of "consistent outperformance".

**Questions:**

Your UEM and UCMR components seem to combine established practices (e.g., CLIP-style triplet learning and score normalization). Could you clarify what specific new insight or mechanism distinguishes CMRAG from prior multimodal retrieval methods?

---

### Official Review · Reviewer_HtZD · 2025-10-30

**Soundness:** 2
**Presentation:** 3
**Contribution:** 2
**Rating:** 4
**Confidence:** 4

**Summary:**

This paper proposes CMRAG, a co-modality RAG framework for visual-document QA that jointly leverages both text and images throughout retrieval and generation, instead of text-only in previous methods. CMRAG includes a Unified Encoding Model (UEM) that embeds queries, parsed document text, and page images into a shared space using a pairwise sigmoid contrastive objective. The extra image encoder and the query encoder are frozen, while text encoder, temperature and bias hyperparameter is updated. CMRAG also proposes a Unified Co-Modality-informed Retrieval (UCMR) that statistically normalizes text- and image-similarity scores before fusion to mitigate distribution mismatch between text and image scores. CMRAG parse page images and document texts with a VLM offline, release a triplet dataset consisting of query, text and image, and achieves consistent gains over single-modality RAG baselines on multiple VDQA benchmarks.

**Strengths:**

**1. Clear unified pipeline**: The paper contrasts text-only and image-only RAG and motivates a unified modality design for retrieval and question answering. UEM reuses SigLIP backbones, freezing query and image encoder, and extending the text encoder for long parsed text, which is practical for indexing and efficient in deployment.

**2. Retrieval score fusion**: Where query and image encoder are frozen, CMRAG sets temperature and
bias hyperparameters learnable and normalizes inner-product scores to a  comparable scale to solve cross-modal score-distribution mismatch. and fuses modalities after putting them on a comparable scale, a clean fix for cross-modal score-distribution mismatch.

**3. Dataset contribution**: The training and evaluation dataset built from existing data is valuable.

**4. Empirical gains**: The method is reported to consistently outperform strong single-modality RAG baselines on multiple retrieval and VDQA benchmarks.

**Weaknesses:**

**1. No adaptation on the visual encoder**:  During UEM training, only text encoder is updated, with query and image encoder frozen, leading the image-side loss do not adapt visual and query encoders. This design may induce alignment issues when images in unseen domains carry information not well captured by the frozen backbones. Could authors show the generalization of CMRAG?

**2. Limited ablations on designs**: Given the architectural choices (freezing query and image encoders; learnable temperature $\tau$ and bias $\eta$ hyperparameters; loss weight $\lambda$ between $\mathcal{L}^T$ and $\mathcal{L}^I$ ), the ablations on them is limited.

**3. Limited novelty**: The retrieval pipeline mainly recombines established ingredients and retrieval paradigm: a joint-embedding setup that projects queries, text, and images into a shared space trained with common pairwise-style losses on top of (largely reused) SigLIP encoders, plus a straightforward statistical normalization to fuse modality scores, which is incremental engineering contribution rather than a fundamentally new retrieval paradigm.

**4. Fairness of comparison**: The reported gains may partly stem from a training-data advantage: the proposed model is trained on a newly constructed in-domain corpus, while key baselines (e.g., SigLIP/SigLIP2) are evaluated from official checkpoints without task-specific fine-tuning. Fine-tune baselines on the same data or add data-matched ablations is necessary.

**Questions:**

Check weaknesses 1-4.

---

### Official Review · Reviewer_i2F8 · 2025-10-31

**Soundness:** 2
**Presentation:** 2
**Contribution:** 2
**Rating:** 2
**Confidence:** 4

**Summary:**

The author classified document retrieval methods into two lines of works: one line of work uses extract text alone for retrieval, while another lines of work treat the document page as image and process using VLM instead. To bridge these two paradigms, the authors propose a Co-Modality-Based RAG framework that combines both textual and visual representations within the retrieval database. Experimental results show that the proposed method outperforms single-modality baselines on visual document question answering benchmarks.

**Strengths:**

- Propose a Co-modality retrieval model that jointly encode the document page with the screenshot and the extracted text.
- Generally better retrieval results on three document retrieval benchmark

**Weaknesses:**

- Missing comparison with multi-token retrievers: Since the goal of this work is to handle document pages where image-based representations may omit fine textual details, a fair comparison should include multi-token (e.g., ColBERT-style) retrievers that explicitly capture token-level interactions to address the same issue as this work. In particular, models such as ColPali or PreFLMR, which are multimodal multi-token retriever, should be considered. Given that ColPali significantly outperforms SigLIP in document retrieval (despite different datasets), and the proposed method also performs better than SigLIP, it would be insightful to compare against these stronger multi-token baselines.
- The evaluation and judgment models appear to come from the same model family, which may introduce bias. Using independent models for evaluation and scoring would strengthen the validity of the results.
- The author should consider experiment on various retrieval backbone to show if the proposed setup performs well regardless what the backbone retriever is. Similarly, the author could experiment with different answer generator, thus the issue of same eval and judge model may also be resolved.
- The paper discusses the impact of the text-query modality weighting but does not present quantitative results. Additional experiments showing how different weights affect retrieval performance would clarify this analysis. Moreover, results demonstrating the effect of normalization would be helpful.
- The proposed setup fix the query encoder and image encoder, while fine-tuning the text document encoder. I was wondering if the image encoder is fine-tuned with the text encoder, i.e., perform end-to-end fine-tuning, would the performance get any better? (Though I understand the author mentioned about preserving multimodal alignment.)

**Questions:**

Regarding document preprocessing with Qwen2.5-VL-7B, have the authors verified the accuracy and consistency of the extracted text and structural information?

---

### Note · Authors · 2025-12-30

I have read and agree with the venue's withdrawal policy on behalf of myself and my co-authors.